# Involvement of Differentially Expressed microRNAs in the PEGylated Liposome Encapsulated ^188^Rhenium-Mediated Suppression of Orthotopic Hypopharyngeal Tumor

**DOI:** 10.3390/molecules25163609

**Published:** 2020-08-08

**Authors:** Bing-Ze Lin, Shen-Ying Wan, Min-Ying Lin, Chih-Hsien Chang, Ting-Wen Chen, Muh-Hwa Yang, Yi-Jang Lee

**Affiliations:** 1Department of Biomedical Imaging and Radiological Sciences, National Yang-Ming University, No. 155, Sec. 2, Linong St. Beitou District, Taipei 11221, Taiwan; innovationmark1012@gmail.com (B.-Z.L.); sywang@mail.femh.org.tw (S.-Y.W.); milo841120@gmail.com (M.-Y.L.); chchang@iner.gov.tw (C.-H.C.); 2Department of Nuclear Medicine, Far Eastern Memorial Hospital, New Taipei City 22000, Taiwan; 3Isotope Application Division, Institute of Nuclear Energy Research, Taoyuan 32546, Taiwan; 4Institute of Bioinformatics and Systems Biology, National Chiao Tung University, Hsinchu 30068, Taiwan; dodochen@nctu.edu.tw; 5Department of Biological Science and Technology, National Chiao Tung University, Hsinchu 30068, Taiwan; 6Center For Intelligent Drug Systems and Smart Bio-devices (IDS2B), National Chiao Tung University, Hsinchu 30068, Taiwan; 7Cancer Progression Research Center, National Yang-Ming University, Taipei 11221, Taiwan; mhyangymu@gmail.com; 8Institute of Clinical Medicine, National Yang-Ming University, Taipei 11221, Taiwan; 9Division of Medical Oncology, Department of Oncology, Taipei Veteran General Hospital, Taipei 11217, Taiwan

**Keywords:** hypopharyngeal cancer, ^188^Re-liposome, repeated therapy, NGS, microRNA

## Abstract

Hypopharyngeal cancer (HPC) accounts for the lowest survival rate among all types of head and neck cancers (HNSCC). However, the therapeutic approach for HPC still needs to be investigated. In this study, a theranostic ^188^Re-liposome was prepared to treat orthotopic HPC tumors and analyze the deregulated microRNA expressive profiles. The therapeutic efficacy of ^188^Re-liposome on HPC tumors was evaluated using bioluminescent imaging followed by next generation sequencing (NGS) analysis, in order to address the deregulated microRNAs and associated signaling pathways. The differentially expressed microRNAs were also confirmed using clinical HNSCC samples and clinical information from The Cancer Genome Atlas (TCGA) database. Repeated doses of ^188^Re-liposome were administrated to tumor-bearing mice, and the tumor growth was apparently suppressed after treatment. For NGS analysis, 13 and 9 microRNAs were respectively up-regulated and down-regulated when the cutoffs of fold change were set to 5. Additionally, miR-206-3p and miR-142-5p represented the highest fold of up-regulation and down-regulation by ^188^Re-liposome, respectively. According to Differentially Expressed MiRNAs in human Cancers (dbDEMC) analysis, most of ^188^Re-liposome up-regulated microRNAs were categorized as tumor suppressors, while down-regulated microRNAs were oncogenic. The KEGG pathway analysis showed that cancer-related pathways and olfactory and taste transduction accounted for the top pathways affected by ^188^Re-liposome. ^188^Re-liposome down-regulated microRNAs, including miR-143, miR-6723, miR-944, and miR-136 were associated with lower survival rates at a high expressive level. ^188^Re-liposome could suppress the HPC tumors in vivo, and the therapeutic efficacy was associated with the deregulation of microRNAs that could be considered as a prognostic factor.

## 1. Introduction

Hypopharyngeal cancer (HPC) represents malignant growth in the hypopharynx region and accounts for about 5% of all head and neck cancers (HNSCC) [1]. As HPC is a rare cancer type with a late occurrence of symptoms and tumor spreading, it is not uncommon for it to be detected at advanced stages with a high mortality rate and poor prognosis [2]. HPC can be treated by conventional surgery, radiotherapy and chemotherapy, while radiotherapy alone is usually used at an early stage [3]. On the other hand, a combination of different therapeutic modalities can improve the five-year survival of this disease [4]. Several lines of evidence have claimed that a combination of radiotherapy and chemotherapy would provide better control of locoregional recurrence compared to surgical procedures [4,5]. A radiopharmaceutical named ^99m^Tc-MIBI (methoxy-isobutyl-isonitrile) has been reported to detect HPC with up to a 95% sensitivity using single photon emission computed tomography (SPECT) [6]. However, nuclear medicine has not been reported to have assessed or monitored the efficacy of HPC therapy as mentioned above. 

Radiopharmaceuticals are not only used for diagnostic purposes but also therapeutic purposes, so-called radiotheranostics [7]. Rhenium-188 (^188^Re) belongs to this type of radionuclide as it emits 85% high-energy β-particles (2.12MeV) and 15% γ-rays (155keV) [8]. The average soft tissue penetration distance of β-particles is only around 3.8mm, suggesting that ^188^Re is suitable for tumor ablation and will not have significant side effects on distant normal tissues [9,10]. ^188^Re has been conjugated to hydroxyethylidene diphosphonate (HEDP) for bone pain palliation [11,12]. The antibody conjugated ^188^Re has also been reported to treat different cancers [13]. Additionally, low immunogenic peptides such as somatostatin derivative conjugated ^188^Re, have been investigated in clinics for patients with advanced pulmonary cancer [14]. ^188^Re-labeled lipiodol and microspheres were used for the treatment of hepatocellular carcinoma [15,16,17]. ^188^Re-labelled radiocolloids have also been developed to treat skin cancers within a brachytherapy device [18]. ^188^Re-loaded lipid nanocapsules, also called ^188^Re-liposome, have been demonstrated to be biocompatible and subjected to a phase 0 clinical study for patients with metastatic tumors [19]. ^188^Re-liposome has shown a theranostic efficacy in various human cancers, including colorectal cancer, glioblastomas, lung cancer, ovarian cancer, esophageal cancer and head and neck cancer using xenograft tumor models [20,21,22,23,24,25,26]. The molecular mechanisms of rhenium-188 labelled radiopharmaceuticals are of interest for investigation aimed at interpreting the potent therapeutic efficacy. 

Radiogenomics is defined as connecting radiomics and genetic profiles to apply the feature of medical imaging in radiation-mediated molecular responses [27]. The purpose of this discipline is to predict the association between gene expression and the radiotherapy-induced toxic effects of tumors [28]. Next-generation sequencing (NGS) analysis is the most cutting-edge technology that can decipher dramatic amounts of gene expressive alterations in tumors, with or without therapies [29,30]. NGS applications include RNA sequencing (RNA-Seq) that can analyze the expression of various RNA populations (mRNA, microRNA, long non-coding RNA, etc.) and their modification forms generated from alternative splicing, mutations or gene fusion [31]. Compared to the conventional cDNA expression microarray, RNA-Seq has a broader spectrum for finding novel and unidentified transcripts [32]. This method is important for providing more detailed and quantitative data for bioinformatics analysis of genetic profiling in novel drug development and predicting the signaling pathways of toxicity and therapeutic effects [33]. As a radioactive compound, ^188^Re-liposome is believed to influence the genetic profile of tumors. However, related studies have rarely been reported. 

In this study, we investigated the effects of ^188^Re-liposome on the miRNA expressive profiles of HPC derived xenograft tumors using RNA-Seq technology. The changed miRNA profiles were analyzed using the Kyoto Encyclopedia of Genes and Genomes (KEGG) pathway database and Ingenuity Pathway Analysis (IPA). The total expressive amount of miRNA from ^188^Re-liposome-treated tumors was about 10% lower than that of untreated tumor. The changed miRNA profile led to 4498 differentially expressed genes (DEGs) that influenced 30 molecular pathways, including olfactory transduction, the cancer pathway, and taste transduction, which accounted for the top three pathways. We also found that ^188^Re-liposome up-regulated several tumor suppressor microRNAs, such as miR-34-5p, miR-193a-5p, miR-125b-5p, miR-133a-5p, and miR-133b-5p. Concomitantly, several oncomirs, including miR-21-5p, miR-32-5p and miR-205-5p were down-regulated by ^188^Re-liposome. An online Kaplan-Meier (K-M) plotter with The Cancer Genome Atlas (TCGA) database was also used to compare the expression of these miRNA and the survival of head and neck cancer patients. The results of ^188^Re-liposome-induced miRNA dysregulation detected by RNA-Seq were discussed.

## 2. Results

### 2.1. Effects of ^188^Re-Liposome on HPC Derived Orthotopic Tumors Using the Repeated Dose Regime

A flowchart of ^188^Re-liposomal manufacturing and the chemical structures of the ^188^Re-perrhenate precursor and BMEDA chelator are illustrated in Figure 1. The timeline was schemed for the establishment of HPC tumor-bearing mice using FaDu-3R cells (Appendix A), the administration of ^188^Re-liposome, the optical imaging of tumor responses, tumor resection for RNA extraction, and NGS analysis (Figure 2A). The growth of orthotopic tumors was significantly suppressed by ^188^Re-liposome but not saline as detected using bioluminescent imaging (Figure 2B,C). The size of resected tumors also exhibited obvious differences between the saline control and ^188^Re-liposome-treated mice after 30 days of initial implantation (Figure 2D). The body weights of tumor-bearing mice were not significantly affected by ^188^Re-liposome (Figure 2E). Furthermore, we showed that DNA damage marker γ-H2AX was significantly up-regulated by ^188^Re-liposome treatment (Figure 2F,G). These data suggest that the regime of ^188^Re-liposome treatment with repeated doses exhibited therapeutic efficacy, with little systemic toxicity.

### 2.2. Use of NGS Analysis to Investigate the microRNA Expressive Profile of HPC Tumor Treated with ^188^Re-Liposome

The resected FaDu HPC tumors treated with saline or repeated doses of ^188^Re-liposome were subjected to RNA extraction to obtain a high quality of total RNA for RNA-seq analysis (Appendix A). The quality of raw data obtained by this analysis was determined by the GC content, by which the raw data quality of the saline control and ^188^Re-liposome-treated tumor was shown to be similar (Appendix A). To confirm the expressive change in microRNA in tumors treated with or without ^188^Re-liposome, the reads number of small RNA (15–55 mers) was extracted and counted using a Small RNA Analysis tool (see Materials and Methods). The average lengths of small RNA after extraction were 32.9 and 32 mers for the control and ^188^Re-liposome treated tumor, respectively. Under this condition, the total reads number of the control and ^188^Re-liposome treated tumor was 10,612,998 and 9,844,775, respectively. The reads were then subjected to the miRBase database (release 21) for the annotation of small RNA using the Annotate and Merge Count of Small RNA Analysis software. The annotated small RNAs of saline-treated tumors and ^188^Re-liposome treated tumors were 639 and 572, respectively. The differentially expressed microRNAs were clustered in a heatmap for a comparison of saline-treated and ^188^Re-liposome treated HPC tumor models (Appendix A). Furthermore, we set the cutoff at five-fold change (log_2_) of microRNA and showed that 13 microRNA and 9 microRNA with mature forms were up-regulated and down-regulated by ^188^Re-liposome normalized to the saline control, respectively (Table 1). The 13 up-regulated microRNAs ranked from highest to lowest fold change were miR-206-3p, mir668-3p, mit-485-3p, miR-382-5p, miR-1268b-5p, miR-193a-5p, miR-7-1-5p, miR-378a-5p, miR-1266-5p, miR-4510-5p, miR-370-3p, miR-34a-5p, and miR-342-5p. The nine down-regulated microRNAs ranked from highest to lowest fold change were miR-142-5p, miR-6723-5p, miR-944-3p, miR-142-3p, miR-136-3p, miR-151b-3p, miR-194-2-5p, miR-143-5p, and miR-3960-3p. According to the online analysis of dbDEMC, ^188^Re-liposome-up-regulated microRNAs were mostly naturally down-regulated in HNSCC and were predicted as tumor suppressors (Table 2). MiR-193a, miR-7-1-5p and miR-342-5p were up-regulated in HNSCC, but could still be tumor suppressors in different types of cancer (see references in Table 2). For the nine ^188^Re-liposome down-regulated microRNAs, only four of them (miR-136-3p, miR-142-3p, miR-944-3p, and miR-142-5p) were reported in the HNSCC of dbDEMC database. Interestingly, three out of these four microRNAs were found to be up-regulated in HNSCC and predicted as oncogenes (Table 3). However, most of the ^188^Re-liposome-down-regulated microRNAs contain oncogenic properties (see references in Table 3). These results suggested that the therapeutic efficacy of ^188^Re-liposome was associated with the deregulation of tumor-suppressive and/or oncogenic microRNAs.

### 2.3. Validation of microRNA Identified in NGS Data Using qPCR

We next used qPCR to validate up-regulated and down-regulated microRNA in the FaDu HPC tumor model treated with ^188^Re-liposome. We selected microRNAs that displayed over five-fold deregulation with the highest RPM, including miR-206-3p, miR-382-5p, miR-378a-5p, miR-3960-3p, and miR-142-5p to be validated. The results showed that the expressive patterns of these microRNAs obtained by qPCR were consistent with the observations of NGS analysis (Figure 3).

### 2.4. Investigation of Differentially Expressed microRNAs in Clinical Samples

As ^188^Re-liposome could influence the expression of certain microRNAs that may correlate with the tumor-suppressive effect of the HPC model, we were interested in examining the status of these microRNAs in clinical HNSCC tumors. First, we obtained clinical HNSCC tissues from patients (n = 6) to investigate the differential expression of microRNAs in tumor tissues and adjacent normal tissues using qPCR analysis. We selected miR-206-3p, miR-378a-5p and miR-142-5p as they exhibited the highest differential deregulation by ^188^Re-liposome (Figure 3). The results showed that the expression of miR-206-3p and miR-378a-5p was down-regulated (Figure 4A,B), while miR-142-5p was up-regulated in tumors compared to normal tissues (Figure 4C). Additionally, we employed the clinical information of HNSCC in the TCGA database to compare the differentially expressed microRNAs in HNSCC and normal tissues. Because the number of cases of hypopharynx cancer was too low to be analyzed, here we used clinical information on larynx cancer types instead. A heatmap was generated for the microRNAs displaying over five-fold change caused by ^188^Re-liposome (Appendix A). Accordingly, we found that miR-206 (equivalent to miR-206-3p) and miR-378a-5p were significantly down-regulated, while miR-143-5p, miR-142-3p, and miR-944 were significantly up-regulated in tumors (Figure 4D). MiR-142-5p also exhibited a trend of up-regulation, although the significance was marginal. This suggests that ^188^Re-liposome can reverse the expression of the microRNAs that were originally deregulated in HNSCC.

### 2.5. Prediction of Genes Targeted by ^188^Re-Liposome-Deregulated microRNAs

We next examined the potent target genes that might be affected by ^188^Re-liposome deregulated microRNAs. The miRDB online database was employed to rank the numbers of predicted genes targeted by microRNAs that were deregulated by ^188^Re-liposome (see Materials and Methods). The ranking of target numbers affected by ^188^Re-liposome up-regulated and down-regulated microRNAs was obtained, respectively (Figure 5A,B). Furthermore, miR-34a, miR-206-3p and miR-4510-5p were used to draw the Venn diagrams as their predicted target genes were ranked as the top three in ^188^Re-liposome up-regulated microRNAs. Only one target, named *SLC44A2* encoded choline transporter-like protein 2 was co-regulated by these three microRNAs (Figure 5C). The same logic was used for ^188^Re-liposome down-regulated microRNAs, and an only one target, named *U2SURP* encoded U2 snRNP-associated SURP motif-containing protein, was co-regulated by miR-142-5p, miR-194-5p and miR-944-3p (Figure 5D). Therefore, these bioinformatics analyses suggest that microRNAs and associated target genes of HPC tumors oppositely regulated by ^188^Re-liposome are distinct.

### 2.6. Analysis of the Molecular Pathways Regulated by ^188^Re-Liposome-Affected microRNA

We next used the pathview R package to investigate the genes influenced by microRNAs regulated by ^188^Re-liposome. MicroRNA samples exhibiting over two-fold change were selected, and the affected target genes were subjected to the KEGG pathway database for determining the potent molecular pathways disturbed by ^188^Re-liposome. We found that thirty pathways in resected HPC tumors were significantly affected by ^188^Re-liposome (*p* < 0.05). The top three pathways with the lowest p values were genes involved in olfactory transduction, pathways in cancer, and taste transduction (Table 4). Additionally, ^188^Re-liposome-influenced pathways could be categorized as cancer and carcinogenesis, cell adhesion and cytoskeletal organization, drug metabolism via cytochrome P450, tumor suppression and oncogenes. An integrated cancer pathway involved in the KEGG database was shown to demonstrate the related genes that could be regulated by ^188^Re-liposome-induced or -suppressed microRNA expression (Figure 5). This revealed that genes associated with cell cycle progression, proliferation, and apoptosis were affected by ^188^Re-liposome, including the down-regulation of *cyclin D*, *cyclin E*, cyclin-dependent kinase (*CDK*) *4/6*, *E2F* transcription factor, and *bcl-2* anti-apoptotic factor, and the up-regulation of *p15*, *p16*, and *p27* cell cycle inhibitors and the *Rb* tumor suppressor gene (Figure 6). This pathway analysis provides a potent profile of the molecular mechanism for ^188^Re-liposome regulated microRNA expression and tumor suppression of the HPC tumor model. 

### 2.7. Association of ^188^Re-Liposome-Regulated microRNA and Patients’ Survival Rate

As ^188^Re-liposome could up-regulate tumor suppressive microRNA or down-regulate oncogenic microRNA of the HPC tumor model, we considered whether these microRNAs could be considered as prognostic factors for patients’ survival rates. The microRNAs deregulated by ^188^Re-liposome that displayed over five-fold change were examined. Notably, the online dataset only includes precursor forms of microRNAs for an analysis of patients’ survival. For ^188^Re-liposome up-regulated microRNAs in the HPC model, only miR-342 and miR-378a potentially exhibited an increase in the survival rate in HNSCC patients, with a marginal significance (Figure 7A,B). Interestingly, a high expression of miR-342 was significantly associated with a better survival rate in female patients (HR = 0.61, 95% CI = 0.38–0.98, *p* = 0.038). On the other hand, ^188^Re-liposome down-regulated microRNAs, including miR-143, miR-6723, miR-944, and miR-136 were associated with a reduced survival rate in HNSCC patients when they were highly expressed (Figure 7C–F). Additionally, miR-3960 was also associated with reduced survival, yet the expression was too low for meaningful analysis (data not shown). These data suggest that specific miRNAs affected by ^188^Re-liposome may be important for perspective clinical evaluation. 

## 3. Discussion

HPC mostly originates from mucosal squamous cells with a low incident rate. Because of unapparent early symptoms and a high metastatic ability, HPC accounts for the lowest survival rate of all head and neck cancers [72]. The FaDu cell line is a squamous cell carcinoma of the human hypopharynx commonly used for the study of molecular mechanisms of head and neck cancer in vitro and in vivo [73]. We have previously established an orthotopic tumor model using this cell line and found that *let-7* microRNA was associated with the therapeutic efficacy of ^188^Re-liposome nanoparticles [24,74]. In this study, we used NGS analysis and found additional microRNAs that might be involved in the therapeutic efficacy of ^188^Re-liposome after repeated administration. The expressions of the total RNA number and annotated small RNA were reduced in ^188^Re-liposome treated tumors, suggesting that ^188^Re-liposome would suppress gene transcription and expression. 

As ^188^Re is a high-energy β particles-emitter, it is expected to induce DNA damage. Indeed, the DNA damage marker γ H2AX was significantly induced by ^188^Re-liposome compared to an untreated control. Interestingly, γ H2AX has been reported to be a tumor suppressor because of its role in the maintenance of genomic stability [75]. In our study, we found that ^188^Re-liposome induced γ H2AX in HPC tumors. In the RNA-seq dataset, we also found that miR-138-2-5p, a potent inhibitor of H2AX [76], was down-regulated by over two-fold by ^188^Re-liposome treatment. Together, these results are partially consistent with previous reports that might account for the therapeutic mechanisms of ^188^Re-liposome from the viewpoint of γ-H2AX-mediated DNA damage responses. 

According to the results of RNA-seq, ^188^Re-liposome induced more than 200 microRNA to change their levels. To raise the selective criteria, we focused on the microRNA species exhibiting over five-fold up-regulation or down-regulation induced by ^188^Re-liposome by comparing them to the untreated controls. The top three up-regulated microRNAs (miR-206-3p, miR-668-3p, and miR-485-3p) and down-regulated microRNAs (miR-142-5p, miR-944-3p, and miR-142-3p) displaying fold-change were categorized as tumor suppressors and oncogenes, respectively (Table 2 and Table 3). Although miR-6723-5p was also highly suppressed by ^188^Re-liposome, its role in cancer has not been interpreted in the dbDEMC database or microRNA Cancer association database (miRCancer) [77]. Most of the microRNAs deregulated by ^188^Re-liposome were expressed oppositely in HNSCC, and they were reported to be potent tumor suppressors or oncogenes in different types of cancers. However, there were no available data for miR-4510-5p, miR-1268b-5p miR-6723-5p, miR-151b-3p, miR-143-5p, miR-194-2-5p and miR3960-3p in the database. As NGS is prominently used to find unknown genes that can be induced by treatment agents, the uncharacterized microRNAs shown to be significantly influenced by ^188^Re-liposome treatment would be of interest for investigating their roles in the future. 

The NGS analysis identified highly differentially expressed microRNAs that were also validated using FaDu tumors with or without ^188^Re-liposome treatment, and clinical HNSCC samples using qPCR. The expression of analyzed microRNAs was consistent in these two different resources, although the case number of clinical samples was limited. The clinical information of the TCGA database could only be employed to analyze larynx cancer because the number of cases of HPC was too low to have normal tissues for analysis. Even so, we still found several microRNAs (e.g., miR-206-3p, miR-378-5p, and miR-142-5p) that were consistent with the results of NGS analysis and our clinical samples (Figure 4). In addition to ^188^Re, ^177^Lu is also a radionuclide that can emit 86% of β-particles and 14% of photons with a lower energy but longer half-life period. ^177^Lu-octreotate has been reported to differentially regulate 57 specific microRNAs in mouse renal cortical tissue identified by the Mouse miRNA Oligo chip 4plex [78]. However, little microRNA was overlapped between their results and ours. Besides different types of radionuclides, the different animal models and microRNA mining methods may also account for the distinct observations in these two studies. Therefore, the differentially expressed microRNAs in the HPC model could be considered as specific prognostic factors for ^188^Re-liposome treatment.

The predicted genes targeted by microRNAs were also analyzed by the miRDB public database. For the ^188^Re-liposome up-regulated and down-regulated top three microRNAs, the *SLC44A2* gene and *U2SURP* gene were the targets recognized by these oppositely regulated microRNAs, respectively. Hence, it is expected that *SLC44A2* would exhibit an oncogenic property and *U2SURP* should be a tumor suppressor gene. *SLC44A2*, first discovered in the inner ear, is a member of the choline transporter-like protein family of membrane transporter proteins [79,80]. *U2SURP* is involved in RNA splicing as it is part of spliceosomes [81]. However, little is known about the association of these two genes with human cancers. It would be an interesting target gene to investigate for its role in mediating the efficacy of ^188^Re-liposome. 

Using the KEGG pathway database, we found that 30 pathways in orthotopic HPC tumors were significantly influenced by ^188^Re-liposome treatment. Although the cancer-suppression-related pathways were expected to be regulated by ^188^Re-liposome, most of the affected genes in the annotated pathway displayed olfactory transduction. Radiation therapy has been reported to cause olfactory loss in head and neck cancer patients [82]. However, the conclusion is that radiation can damage olfactory cells. The position of hypopharyngeal cancer was not in the olfactory tract, yet FaDu cells were orthotopically injected into the buccal position of the mouse. This operation was based on the fact that FaDu cells are also buccal carcinoma cells [83]. Whether the microenvironmental difference influences the gene regulatory pathway of tumors is unclear. We believed that the excised tumor should not be contaminated by olfactory cells because the human tumor was very small after ^188^Re-liposome treatment, that is, the tumor size was not big enough to reach the olfactory tract. 

An assessment of the correlation between the gene expression and survival rate of patients is important for evaluating the clinical relevance of preclinical study for novel genes and drugs. Here we used the Kaplan–Meier plotter online tool that includes the datasets of TCGA program, the Gene Expression Omnibus (GEO) and the European Genome-Phenome Archive (EGA) to find the ^188^Re-liposome regulated primary microRNAs and their association with patients’ survival rates [84]. Although several potent tumor suppressive or oncogenic microRNAs were influenced by ^188^Re-liposome, only part of these microRNAs exhibited the expected association with patients’ survival probability in HNSCC. For instance, ^188^Re-liposome suppressed miR-142 exhibited higher survival probabilities in HNSCC when they were expressed at higher levels, although they were expected to be oncogenic microRNAs [85]. A potent limitation is the sample size (523 cases) of HNSCC patients, which may be too small to draw conclusions on the role of microRNAs in patients’ survival rates. Besides, the online K-M plotter only analyzes the effects of precursor microRNA on patients’ survival rate. The association of mature miRNAs with the survival rate remains unknown. Whether different forms of the same microRNAs will differentially influence the results of the survival rate remains to be addressed. 

In summary, current data suggest that the de-regulation of microRNAs correlates with the therapeutic efficacy of ^188^Re-liposome on human HPC tumors. Using NGS, we also found several microRNAs that have not been fully characterized for their roles in cancer development and therapy. Whether these microRNAs are important for mediating the efficacy of ^188^Re-liposome would be interesting to further investigate. Additionally, the KEGG pathway analysis showed that not only cancer pathways but also olfactory and taste transduction were significantly changed in HPC tumors after they were treated by ^188^Re-liposome. Although no clinical evidence showing that patients treated with ^188^Re-liposome will lose olfactory and gustatory sensation, olfactory sensory dysfunction and gustatory impairment often occur after patients are treated with radiotherapy in the head and neck area [82,86,87,88,89]. To the best of our knowledge, this is the first study uncovering the therapeutic mechanisms of ^188^Re-liposome by an investigation of the pan-expression of microRNA. As ^188^Re-liposome has entered the clinical trial stage, these data may further extend the concept of precise medicine using this radiotheranostic agent and allow the affected microRNAs to be prognostic factors after cancer treatment. 

## 4. Materials and Methods

### 4.1. Cell Lines and Plasmid

Human FaDu HPC cells (American Type Culture Collection, Manassas, VA, USA) were cultured in RPMI-1640 (Life Technologies Inc., Carlsbad, CA, USA) containing 10% fetal bovine serum (FBS) (Thermo Fisher Scientific Inc., Waltham, MA, USA), 1% penicillin (Sigma-Aldrich Co., St. Louis, MO, USA), and 1% L-glutamine (Sigma-Aldrich Co., St. Louis, MO, USA). FaDu-3R cells harboring multiple reporter genes were used and cultured as reported previously [24]. Cells were incubated at 37^o^C in a humidified incubator with 5% CO_2_ and passaged every two days.

### 4.2. Preparation of ^188^Re-Liposome

The procedure of ^188^Re-liposomal preparation has been reported before [23]. In brief, ^188^Re was milked from the ^188^W/^188^Re generator system (Institute National des Radioelements, Fleurus, Belgium) and conjugated with sodium perrhenate. Moreover, ^188^Re was conjugated with *N*,*N*-bis(2-mercapatoethly)-*N*′,*N*′-diethylenediamine (BMEDA, ABX GmbH, Radeberg, Germany), and the quality of ^188^Re-BMEDA was validated by using the instant thin-layer chromatography (iTLC) followed by a radioactive scanner (Bioscan AR2000; Bioscan, TriFoil Imaging Inc., Chatsworth, CA, USA). Furthermore, PEGylated liposome (NanoX; Taiwan Liposome Co. Ltd., Taipei, Taiwan) was used to encapsulate ^188^Re-BMEDA and eluted using the PD-10 column (GE Health BioSciences, Pittsburgh, PA, USA) (Appendix A). The average molecular weight of polyethylene glycol (PEG) was 2000. The particle size (84.6 ± 4.12 nm) and surface charge (1.1 ± 1.9 mV) were measured by the dynamic light scattering apparatus (Zetasizer Nano ZS90, Malvern Panalytical Ltd., Malvern, UK). The in vitro stabilities of ^188^Re-liposome in normal saline and rat plasma were, respectively, over 92% and 82% in 72 h as reported before [20].

### 4.3. Establishment of HPC Orthotopic Tumor Model for Evaluating the Therapeutic Efficacy of ^188^Re-Liposome

Six-week-old male BALB/c nude mice (*N* = 5 for each experimental group) were purchased from National Laboratory Animal Center, Taipei, Taiwan and used for the establishment of orthotopic HPC tumor model. FaDu-3R cells (1 × 10^6^) were resuspended in 50 μL of OPTI-MEM (Sigma-Aldrich, St. Louis, MO, USA) and then injected into the buccal position of each mouse at right side using a 27 G insulin needle. For intravenous injection of ^188^Re-liposome, 23.68 MBq (640 μCi) corresponding to 80% maximum tolerated dose (MTD), as we mentioned before [23]. To evaluate the therapeutic efficacy, the tumor viability and growth rate were measured using the luciferase reporter gene imaging and caliper measurement. The luminescent signals were acquired by the In Vivo Imaging System (*Optima*, Biospace Lab Inc., Paris, France). The tumor volume was calculated by the formula: (width^2^ × length)/2 after caliper measurement every three days [90]. The animal experiments were approved by the Institutional Animal Care and Utilization Committee (IACUC) of National Yang-Ming University (No. 1061010).

### 4.4. Tumor Collection and Next-Generation Sequencing (NGS)

Tumors were harvested from tumor-bearing mice after four weeks of ^188^Re-liposome treatment. Total RNA of both saline control and ^188^Re-liposome treated group were extracted using the QIAGEN RNA mini kit (Thermo Fisher Scientific Inc., Waltham, MA, USA) according to manufacturer’s instructions. Furthermore, the quality of RNA was detected using the Nanodrop spectrophotometer (Nanodrop Technologies LLC, Wilmington, DE, USA). The integrity and concentration of RNA samples were determined using the Agilent 2100 Bioanalyzer (Agilent Technologies, Santa Clara, CA, USA) with RNA 6000 nano kit (Agilent Technologies, Santa Clara, CA, USA). The TruSeq Small RNA Library kit (Illumina, Inc., San Diego, CA, USA) was then used to ligate RNA with adapters followed by the reverse transcription-PCR to generate cDNA library. The library was then sequenced by the HiSeq 4000 Sequencing System (2 × 150 bp paired-end Sequencing) and the results were processed with the Illumina software (Illumina, Inc., San Diego, CA, USA). To qualify the results of small RNA sequencing, the sequences were applied to the CLC Genomics Workbench v10 to obtain the qualified reads [91]. CLC Genomics Workbench counts different types of small RNAs in the data and compares them to databases of microRNAs or other small RNAs [92].

### 4.5. MicroRNA Expression Analysis

The ‘miRBase’ online source was used for the annotation of small RNA [93]. In the next step, all of the miRNAs were normalized by TMM (trimmed mean of M values) method using edgeR (R package: v.3.10.5) (Bioconductor, New York City, NY, USA) [94]. For a two-group experiment, we used the ‘Fold Change’ to tell how many times bigger the mean expression value in ^188^Re-liposome group is relative to that of saline-only group. If the mean expression value in ^188^Re-liposome group is small than that in saline-only group, the value will present negative sign, and vice versa. The criteria for microRNAs selection were fold change >5.

### 4.6. Western Blot Analysis

Tumors were collected from the tumor-bearing mice after 4 weeks of ^188^Re-liposome treatment, and lysed in T-PER^TM^ Tissue Protein Extraction Reagent (Thermo Fisher Scientific, Waltham, MA, USA) containing 1% protease inhibitor cocktail (Sigma-Aldrich Co., St. Louis, MO, USA). Protein lysates (50 μg) were run on 8–12% SDS-PAGE, electro-transferred to nitrocellulose membrane, blocking and incubated with antibody as reported previously [95]. The primary antibodies were anti-γH2AX (GTX628789, GeneTex Inc., Irvine, CA, USA), and anti-glyceraldehyde3-phosphate dehydrogenase (GAPDH) (MA5-15738, Invitrogen Inc., Carlsbad, CA, USA).

### 4.7. Validation of microRNA Expression Using qPCR

To validate miRNA levels before and after HPC tumor treated with ^188^Re-liposome and to compare normal tissue to HNSCC tissues using clinical samples, quantitative PCR (qPCR) of targeted miRNA was performed. Briefly, complementary DNA (cDNA) was generated from 2 μg total RNA using SuperScript II reverse transcriptase (Life-Technologies Co., Carlsbad, CA, USA). Then, the cDNA products were mixed with the Fast SYBR Green Master Mix (Life-Technologies Co., Carlsbad, CA, USA) and subjected to the StepOnePlus Real-time PCR System (Life-Technologies Co., Carlsbad, CA, USA) following the manufacturer’s instructions. The sequences of stem loop primers, forward primers and universal primer used for miR-206-3p, miR-382-5p, miR-378a-5p, miR-3960-3p, and miR-142-5p amplification were summarized in Table 5. Use of human tissue samples was approved by the Institutional Review Board (No. 2019-01-010BC).

### 4.8. Heatmap Analysis of NGS Data

To gain the heatmap of total microRNAs’ expression, ‘TreeView’ (v1.1.6r4) was used [96]. The heatmap analysis was based on the small RNA that has twice difference between individuals with or without the treatment of ^188^Re-liposome, and then took Log2 value to draw out.

### 4.9. Analysis of microRNA Using the Cancer Genome Atlas (TCGA)

The expression levels of miRNAs and clinical information for TCGA Head-Neck Squamous Cell Carcinoma (HNSC) were downloaded from Broad GDAC Firehose [97]. The miRNA expression values for samples having anatomic subdivision labeled as normal larynx tissue (12 cases), larynx cancers (117 cases) or hypopharynx cancers (10 cases) were used. In house R scripts were used to parse and generate heatmap and boxplots [98]. The difference between normal and tumor samples were tested with Wilcoxon signed-rank test.

### 4.10. Characterization of miRNAs

The miRNA of interests were subjected to a Database of Differentially Expressed MiRNAs in human Cancer 2.0 (dbDEMC v2.0) that collects the data sets of Gene Expression Omnibus (GEO) and The Cancer Genome Atlas (TCGA) to exhibit differentially expressed miRNAs in human cancers detected by high throughput method [99]. The roles of miRNAs belonging to tumor suppressor genes or oncogenes were predicted accordingly. The miRDB online database for prediction of miRNA targets by a MirTarget bioinformatics tool was used to analyze putative downstream genes influenced by the miRNAs of interests [100,101]. Each microRNA was selected for target expression analysis for FaDu cells, and the targets expression level over 20 was counted as they were most relevant to FaDu cells. Target expression level was determined by RNA-Seq using the RPKM method (Reads Per Kilobase of transcript, per Million mapped reads). An online Venn diagrams drawing tool was exploited to calculate the intersections of list of miRNA targets [102].

### 4.11. The Pathway Analysis

The ‘pathview’ (R package: v1.4.2) software (Bioconductor, New York City, NY, USA) was used to draw pathways and find significant gene change (Fold change > 2) with the Kyoto Encyclopedia of Genes and Genomes (KEGG) [103,104]. Moreover, the pathways would be regarded significant if the *p*-value < 0.05.

### 4.12. Statistical Analysis

Statistical analysis was performed using GraphPad Prism 6.0 (GraphPad Software, San Diego, CA, USA). All data were represented as the means ± standard deviation (SD) with independent experiments. The Student’s *t*-test was used for statistical analysis. Two-way Analysis of Variance (ANOVA) was used to compare the tumor growth curves. The Kaplan–Meier method with the log-rank test was used to analyze the association of miRNAs and patients’ survival rates using the online K-M plotter with public datasets [84].

## Figures and Tables

**Figure 1 molecules-25-03609-f001:**
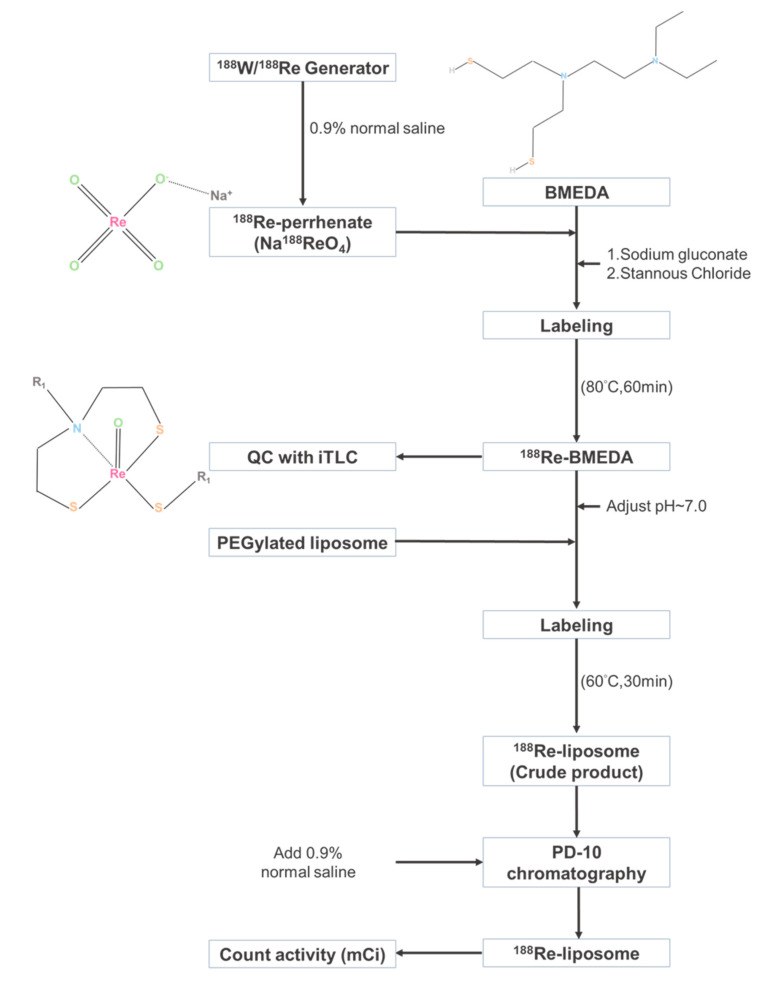
The flowchart of ^188^Re-liposomal preparation. The chemical structures of Na^188^ReO_4_, BMEDA and formed ^188^Re-BMEDA were also illustrated.

**Figure 2 molecules-25-03609-f002:**
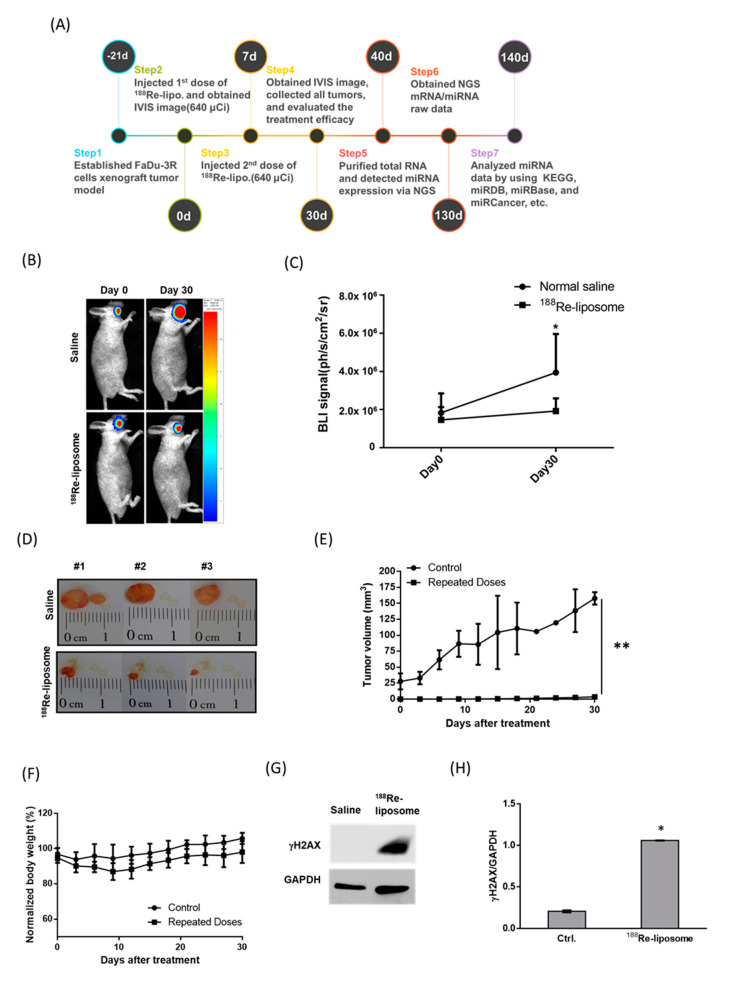
Comparison of PEGylated ^188^Re-liposomal accumulation in orthotopic hypopharyngeal cancer (HPC) tumors after repeated injections. (**A**) The experimental scheme for ^188^Re-liposome treatment and analysis. (**B**) Reporter gene imaging of tumor growth responding to repeated doses of ^188^Re-liposome, and the saline-treated control. (**C**) Quantification of bioluminescent imaging (BLI) signals. *: *p* < 0.05. (**D**) Representative photos of excised orthotropic tumors with or without the treatment of ^188^Re-liposomes. (**E**) Caliper measurement of tumor volumes. Data are represented as means ± S.D. **: *p* < 0.01. (**F**) Measurement of body weights of mice. (**G**) Comparison of the γ-H2AX protein expression in tumors with or without the treatment of ^188^Re-liposomes. (**H**) Densitometric quantification of Western blots. *: *p* < 0.05.

**Figure 3 molecules-25-03609-f003:**
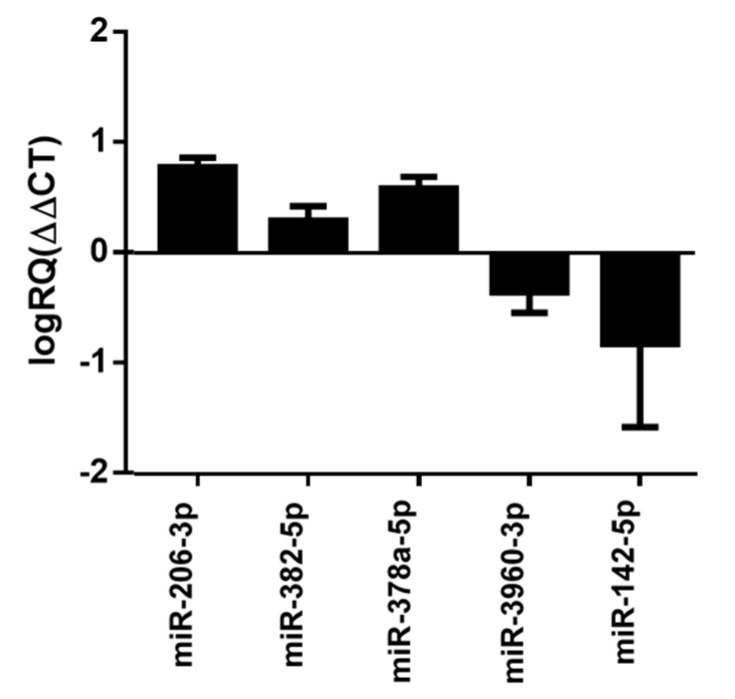
Validation of the next-generation sequencing (NGS) results of the HPC tumor model by qPCR analysis. The results were each microRNA of ^188^Re-liposome-treated HPC normalized to that of untreated controls.

**Figure 4 molecules-25-03609-f004:**
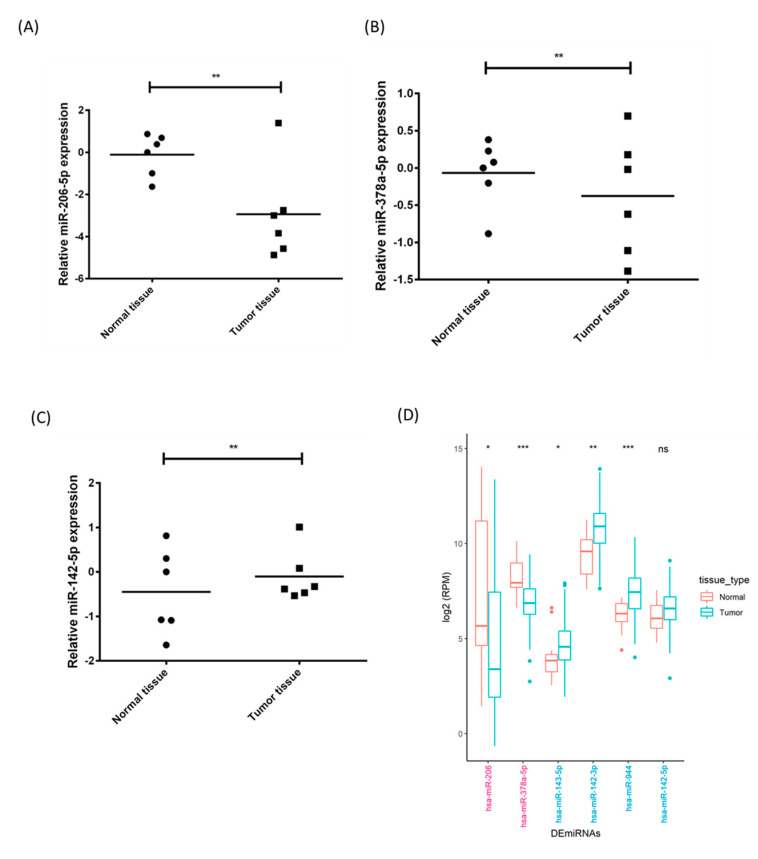
Comparison of the microRNA expression in clinical head and neck cancer (HNSCC) tissues and adjacent normal tissues by qPCR analysis. (**A**) MiR-206-3p. (**B**) MiR-378a-5p (**C**) MiR-142-5p. (**D**) Differentially expressed microRNA (DEmiRNA) in larynx tumor and normal tissues using the clinical information of The Cancer Genome Atlas (TCGA). *: *p* < 0.05, **: *p* < 0.01. ***: *p* < 0.001.

**Figure 5 molecules-25-03609-f005:**
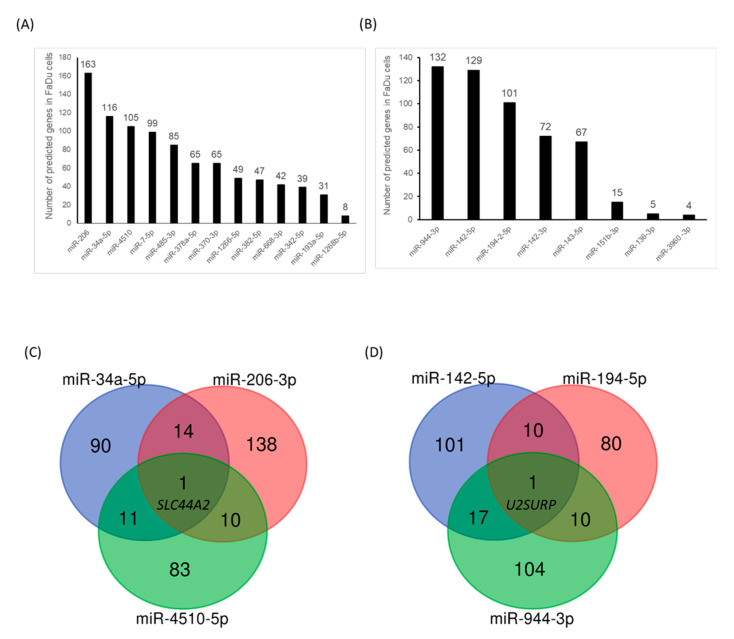
Analysis of miRNA-target interactions. (**A**) The individual target number of microRNAs up-regulated by ^188^Re-liposome. (**B**) The individual target number of microRNAs down-regulated by ^188^Re-liposome. (**C**,**D**) The Venn diagram calculated and drawn by the three microRNAs with the most targets.

**Figure 6 molecules-25-03609-f006:**
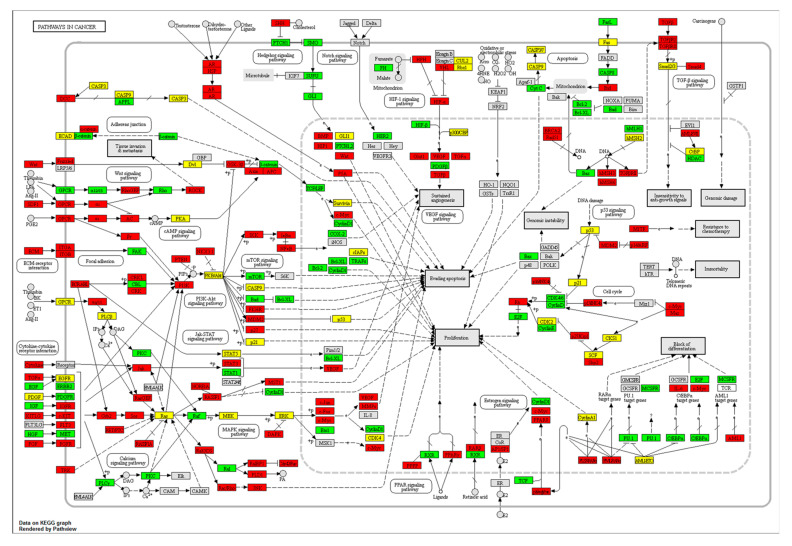
Use of the Kyoto Encyclopedia of Genes and Genomes (KEGG) database to display cancer pathway-associated genes affected by ^188^Re-liposome mediated miRNA expression in HPC. Red: Up-regulated genes; green: down-regulated genes; yellow: unknown regulated genes; gray: unchanged genes.

**Figure 7 molecules-25-03609-f007:**
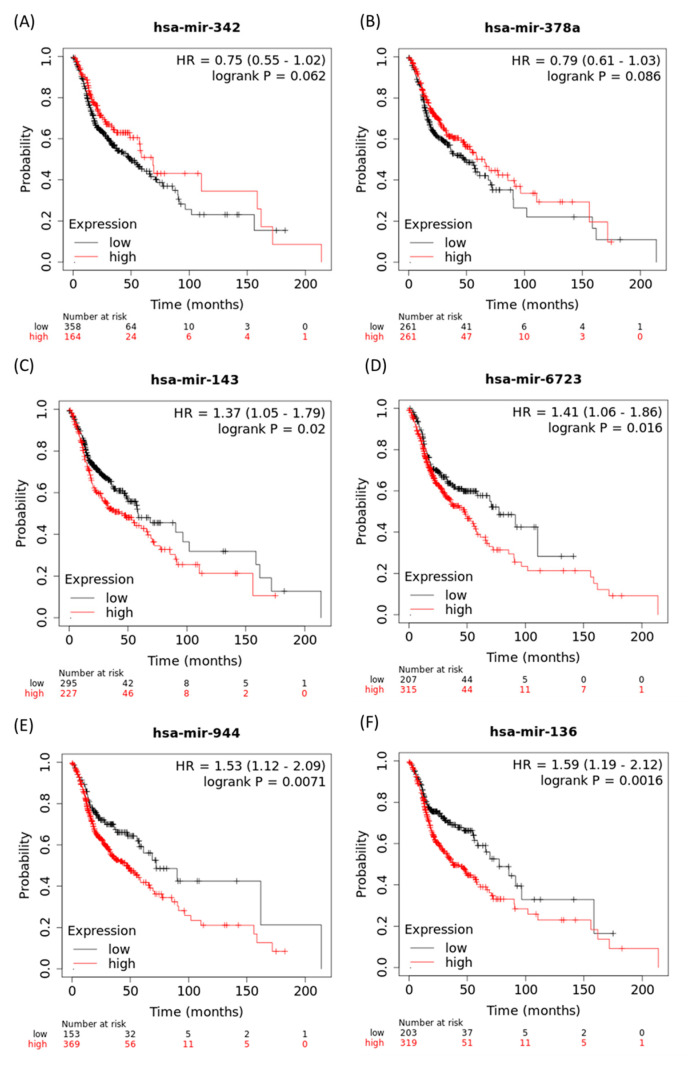
The association of ^188^Re-liposome deregulated microRNAs with the survival rates of HNSCC patients. Kaplan–Meier (K-M) plot (**A**) hsa-miR-342; (**B**) hsa-miR-378a; (**C**) hsa-miR-143; (**D**) hsa-miR-6723; (**E**) hsa-miR-944; (**F**) hsa-miR-136.

**Table 1 molecules-25-03609-t001:** Expression of microRNAs in human hypopharyngeal tumor model treated with 188Re-liposome.

MicroRNA	Fold Change (188Re-lipo./Ctrl.) ^a^	Ctrl. Norm. by TMM ^b^	^188^Re-lipo. Norm.by TMM	Ctrl. RPM ^c^	^188^Re-lipo. RPM
Hsa-miR-206-3p	40.19	76.05	3056.53	2.92	90.71
Hsa-miR-668-3p	11.16	2.45	27.38	0.09	0.81
Hsa-miR-485-3p	9.77	2.45	23.96	0.09	0.71
Hsa-miR-382-5p	9.15	22.08	201.94	0.85	5.99
Hsa-miR-1268b-5p	8.37	2.45	20.54	0.09	0.61
Hsa-miR-193a-5p	6.28	4.91	30.8	0.19	0.91
Hsa-miR-7-1-5p	6.28	9.81	61.61	0.38	1.83
Hsa-miR-378a-5p	5.78	51.52	297.78	1.98	8.84
Hsa-miR-1266-5p	5.58	2.45	13.69	0.09	0.41
Hsa-miR-4510-5p	5.58	2.45	13.69	0.09	0.41
Hsa-miR-370-3p	5.58	4.91	27.38	0.19	0.81
Hsa-miR-34a-5p	5.15	127.57	657.17	4.9	19.5
Hsa-miR-342-5p	5.12	7.36	37.65	0.28	1.12
Hsa-miR-3960-3p	−5.73	117.76	20.54	4.52	0.61
Hsa-miR-143-5p	−5.73	19.63	3.42	0.75	0.1
Hsa-miR-194-2-5p	−5.73	19.63	3.42	0.75	0.1
Hsa-miR-151b-3p	−6.45	22.08	3.42	0.85	0.1
Hsa-miR-136-3p	−7.17	24.53	3.42	0.94	0.1
Hsa-miR-142-3p	−7.88	26.99	3.42	1.04	0.1
Hsa-miR-944-3p	−9.32	31.89	3.42	1.22	0.1
Hsa-miR-6723-5p	−10.03	34.35	3.42	1.32	0.1
Hsa-miR-142-5p	−10.14	242.88	23.96	9.33	0.7

^a^—Fold change in microRNA over 5 or below −5 were selected. ^b^—TMM: Trimmed mean of M values. ^c^—RPM: Reads of exon model per million mapped reads. (ExonMappedReads × 10^6^/TotalMapped Reads).

**Table 2 molecules-25-03609-t002:** Functional prediction algorithm for microRNAs of human hypopharyngeal tumor model up-regulated by ^188^Re-liposome.

MiRNAs up-Regulated in HPC	Original Change in HNSCC ^a^	Role Prediction	Functional Importance	GEO ID	References ^b^
Hsa-miR-206-3p	down-regulation	Tumor suppressor	Weaks cell proliferation, migration, invasion, promotes S phase cell arrest. Inhibits cell aggressiveness	TCGA_HNSC	[34,35]
Hsa-miR-668-3p	down-regulation	Tumor suppressor	Induces growth arrest and premature senescence	- ^c^	[36,37]
Hsa-miR-485-3p	down-regulation	Tumor suppressor	Inhibits mitochondrial biogenesis or promotes cancer growth and migration	GSE75630	[38,39,40]
Hsa-miR-382-5p	down-regulation	Tumor Suppressor	Promotes lymph node metastasis and TNM stage; inhibition of proliferation and EMT in glioma cells	TCGA_HNSC	[41,42]
Hsa-miR-1268b-5p	-	Tumor suppressor	Increases chemosensitivity	-	[43]
Hsa-miR-193a-5p	up-regulation	Tumor Suppressor	Suppresses the growth and the metastasis of cancer cells	TCGA_HNSC	[44,45]
Hsa-miR-7-1-5p	up-regulation	Tumor suppressor	Inhibits proliferation, invasion and induces apoptosis in cancer cells	TCGA_HNSC	[46,47]
Hsa-miR-378a-5p	down-regulation	Tumor suppressor	Inhibits cellular proliferation and colony formation.	TCGA_HNSC	[48,49]
Hsa-miR-1266-5p	down-regulation	Tumor suppressor	Induces apoptosis and reduces proliferation	TCGA_HNSC	[50,51]
Hsa-miR-4510-5p	-	Tumor suppressor	Down-regulation in recurrent cancer and a potential cancer biomarker	-	[52,53]
Hsa-miR-370-3p	down-regulation	Tumor suppressor	Potential cancer biomarker	TCGA_HNSC	[54]
Hsa-miR-34a-5p	down-regulation	Tumor suppressor	Inhibits tumorigenesis and progression	TCGA_HNSC	[55,56]
Hsa-miR-342-5p	up-regulation	Tumor suppressor	Reduces cell cycle progression	TCGA_HNSC	[57,58]

^a^—The change of microRNA was determined by the dbDEMC online databases. ^b^—The selected references may be not HPC or HNSCC related. ^c^—The tumor suppressive function was concluded from clinical patients.

**Table 3 molecules-25-03609-t003:** Target prediction algorithm for microRNAs of human hypopharyngeal tumor model down-regulated by ^188^Re-liposome.

MiRNAs Down-Regulated in HPC	Original Change in HNSCC	Role Prediction	Functional Importance	GEO ID	Reference
Hsa-miR-3960-3p	-	-	-	-	-
Hsa-miR-143-5p	-	Both	cell viability, colony formation	-	[59,60]
Hsa-miR-194-2-5p	-	Oncogene	cell proliferation, migration and invasion	-	[61,62]
Hsa-miR-151b-3p	-	Oncogene? ^a^	Biomarker of sarcoma	-	[63]
Hsa-miR-136-3p	down-regulation	Oncogene	Biomarker of bladder cancer, promote cancer growth and migration	TCGA-HNSC	[64,65]
Hsa-miR-142-3p	up-regulation	Oncogene	Over-expression in OSCC, association with cancer growth and migration	TCGA-HNSC	[66,67]
Hsa-miR-944-3p	up-regulation	Oncogene	A biomarker of poor prognosis/Regulation of chemoresistance	TCGA-HNSC	[68,69]
Hsa-miR-6723-5p	-	-	-	-	-
Hsa-miR-142-5p	up-regulation	Oncogene	Deregulation of cell proliferation; SMAD3/TGF-β	GSE31277	[70,71]

^a^ No direct evidence, but just an implication.

**Table 4 molecules-25-03609-t004:** Thirty significant changed pathways after repeated doses of ^188^Re-liposome treatment.

Pathways Description	No. of DEGs with Annotated Pathways (4498) ^a^	Percentage of DEGs with Annotated Pathways (4498)	Down Regulated Gene	Up Regulated Gene	Unknown Regulated Gene	No. of All Genes with Annotated Pathways (6883)	Percentage of All Genes with Annotated Pathways (6883)	*p*-Value ^b^
Olfactory transduction	51	1.13%	14	33	4	415	6.03%	6.37 × 10^−45^
Pathways in cancer	329	7.31%	95	184	50	397	5.77%	0.00127
Taste transduction	16	0.36%	2	11	3	52	0.76%	0.00373
HTLV-I infection	217	4.82%	61	117	39	258	3.75%	0.00623
Proteoglycans in cancer	176	3.91%	60	93	23	203	2.95%	0.00739
Neuroactive ligand-receptor interaction	139	3.09%	40	84	15	277	4.02%	0.00779
MicroRNAs in cancer	151	3.36%	41	89	21	297	4.31%	0.00885
Viral carcinogenesis	175	3.89%	47	90	38	205	2.98%	0.0103
Chemical carcinogenesis	32	0.71%	4	21	7	81	1.18%	0.01124
Hippo signaling pathway	135	3.00%	34	82	19	154	2.24%	0.01459
Focal adhesion	174	3.87%	55	90	29	207	3.01%	0.01626
MAPK signaling pathway	209	4.65%	53	128	28	255	3.70%	0.01732
Drug metabolism - cytochrome P450	27	0.60%	5	17	5	68	0.99%	0.01947
Metabolism of xenobiotics by cytochrome P450	30	0.67%	4	20	6	74	1.08%	0.01954
ErbB signaling pathway	82	1.82%	30	40	12	87	1.26%	0.02121
Signaling pathways regulating pluripotency of stem cells	124	2.76%	29	80	15	142	2.06%	0.02202
Endocytosis	208	4.62%	65	116	27	258	3.75%	0.02587
FoxO signaling pathway	117	2.60%	32	75	10	134	1.95%	0.02612
Ras signaling pathway	185	4.11%	50	109	26	227	3.30%	0.0272
Neurotrophin signaling pathway	106	2.36%	28	64	14	120	1.74%	0.02764
Regulation of actin cytoskeleton	175	3.89%	49	98	28	214	3.11%	0.03031
Chronic myeloid leukemia	70	1.56%	22	40	8	73	1.06%	0.03053
Transcriptional misregulation in cancer	149	3.31%	43	80	26	179	2.60%	0.03365
Glioma	63	1.40%	23	30	10	65	0.94%	0.03554
Pancreatic cancer	64	1.42%	19	35	10	66	0.96%	0.03682
Colorectal cancer	60	1.33%	17	30	13	62	0.90%	0.03971
Acute myeloid leukemia	56	1.24%	17	28	11	57	0.83%	0.04123
Protein processing in endoplasmic reticulum	140	3.11%	40	78	22	169	2.46%	0.04447
Prostate cancer	80	1.78%	26	38	16	89	1.29%	0.04703
TGF-beta signaling pathway	76	1.69%	23	46	7	84	1.22%	0.04999

^a^ DEG: Differentially expressed genes. ^b^ Only *p* < 0.05 was counted.

**Table 5 molecules-25-03609-t005:** The list of stem loop primers, forward primers, and reverse primer used for qPCR of microRNAs.

**MicroRNA**	**Stem Loop Primer Sequence**
miR-206-3p	5′-GTCGTATCCAGTGCAGGGTCCGAG
	GTATTCGCACTGGATACGACCCACAC-3′
miR-382-5p	5′-GTCGTATCCAGTGCAGGGTCCGA
	GGTATTCGCACTGGATACGACCGAATC-3′
miR-378a-5p	5′-GTCGTATCCAGTGCAGGGTCCGAG
	GTATTCGCACTGGATACGACACACAG-3′
miR-3960-3p	5′-GTCGTATCCAGTGCAGGGTCCGAG
	GTATTCGCACTGGATACGACCCCCCG-3′
miR-142-5p	5′-GTCGTATCCAGTGCAGGGTCCGAG
	GTATTCGCACTGGATACGACAGTAGT-3′
**MicroRNA**	**Forward Primer Sequence**
miR-206-3p	5′-CACGCATGGAATGTAAGGAAGT-3′
miR-382-5p	5′-CACGCAGAAGTTGTTCGTGGTG-3′
miR-378a-5p	5′-TGATTACTCCTGACTCCAGGTC-3′
miR-3960-3p	5′-TAATTATGGCGGCGGCGGAG-3′
miR-142-5p	5′-CACGCGCATAAAGTAGAAAGCA-3′
**MicroRNA**	**Reverse Primer Sequence**
Universal reverse primer	5′-CCAGTGCAGGGTCCGAGGT-3′

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
