# Peer review of "Involvement of Differentially Expressed microRNAs in the PEGylated Liposome Encapsulated 188Rhenium-Mediated Suppression of Orthotopic Hypopharyngeal Tumor"

_molecules, 2020, doi:10.3390/molecules25163609_

Round 1

Reviewer 1 Report

The manuscript by Lin et al. deals with the differential expressed microRNAs involved with the therapeutic efficacy of 188Re-containing liposomes, which are currently also being investigated in clinical trials. The approach of studying the deregulated microRNAs to further elucidate the therapeutic pathway is an interesting one. The authors have studied the problem in a proper manner, using appropriate methodology, and the manuscript can be considered for publication in Molecules after some key concerns are addressed in a revised form of the submission. 

It is highly recommended that the authors go through the manuscript to correct the English grammar issues, or ask a native speaker to do so. A few have been highlighted below in the introduction section, but there are quite a few more in all sections. Underneath all the figures it says ‘Fig X Lin et al.’ and then still the caption. Please remove the “Fig X Lin et al.’ text. Furthermore, some suggestions for improvement are given per manuscript section:

Abstract

  • Line 5: “HPC tumor” should be “HPC tumors”

Introduction

  • Line 4: “High mortaliy rate” (not mortal rate)
  • Line 9: Ti is not the element, I think you mean Tl here.
  • 2nd paragraph “Radiopharmaceutical is” should be “radiopharmaceuticals are” and “diagnostic” and “therapeutic purposes” (instead of purpose).
  • There are many other instances in which the English should be improved, I did not write them down but please have the manuscript checked.

Methods

  • Under Preparation of 188Re liposome some information concerning the methods is missing. Which lipid did you use specifically? Either state the liposome preparation and 188Re encapsulation method here, or refer to one of your earlier papers for full details. What DLS apparatus (company name?) did you use?
  • Under Establishment of HPC Orthotopic Tumor Model for Evaluating the Therapeutic Efficacy of 188Re-liposomeyou mention that 5 mice were purchased for the experiment. However, later on in the paper you also mention a saline control group. Was the entire experiment only based on 5 mice in total?
  • In the same section, you calculated tumor volume using (w^2 x l)/2. This appears that you did not assume a spherical tumor, but rather a rectangular one?
  • In the methods section, direct website links are included. Please replace these by proper references.
  • The section Validation of microRNA expression using qPCR contains some very long sequences. It is probably easier to read if this information can be presented in a table.

Results

  • Figure 1 provides a very nice overview of the obtained results. However, it would look even better if all the graphs had the same style (e.g. font and font size)
  • At the moment it is not possible to read the information in Figure 5. A better quality figure would be great. Also for figure 6 a better quality image of the graphs would be appreciated.
  • Table 1 currently the caption is on a different page than the table itself. Next to that, underneath the table the meaning of a, b and c are explained, but I cannot find “a” in the table

Discussion

  • You mention that the liposomes should not only influence tumor growth but also olfactory sensation and gustatory sensation. Was any clinical evidence found in the patients treated with 188Re which corroborates this?
  • In the discussion, it could be interesting to compare the results from your study with other studies where they looked at microRNA expression after radionuclide administration (e.g. https://doi.org/10.1371/journal.pone.0112645)

Author Response

Dear reviewer,

Sincerely,

Yi-Jang Lee

Reviewer 2 Report

Lin et al., reported the results of animal studies on the therapeutic effects and mechanisms of 188Re-liposome by investigation of pan-expression of microRNA. 

This paper may be improved after the following revisions.

  1. Please add a scheme showing the procedure of radiolabeling reaction including the chemical structure (e.g. chelator)
  2. Authors should add in vitro stability data of 188Re-labeled liposome in mouse serum to check whether the radioisotope can be liberated from the liposome in biological media.
  3. Please indicate the size of PEG on the liposome. 
  4. The text and abbreviation in Figure 5 is too small and blurry to read; please revise it. 
  5. Delete fig..x Lin et al. in the figures (not figure legend) pages 6, 10, 11, and 17.    

Author Response

Dear reviewer,

Sincerely,

Yi-Jang Lee

Round 2

Reviewer 2 Report

The revised one is improved (figures and presentation of results) compared with the original manuscript. 

I do not have further comments for this paper.